# Face Image Segmentation Using Boosted Grey Wolf Optimizer

**DOI:** 10.3390/biomimetics8060484

**Published:** 2023-10-12

**Authors:** Hongliang Zhang, Zhennao Cai, Lei Xiao, Ali Asghar Heidari, Huiling Chen, Dong Zhao, Shuihua Wang, Yudong Zhang

**Affiliations:** 1Jilin Agricultural University Library, Jilin Agricultural University, Changchun 130118, China; zhl@jlau.edu.cn; 2College of Computer Science and Artificial Intelligence, Wenzhou University, Wenzhou 325035, China; cznao@wzu.edu.cn (Z.C.); xiaolei@wzu.edu.cn (L.X.); 3School of Surveying and Geospatial Engineering, College of Engineering, University of Tehran, Tehran 11366, Iran; as_heidari@ut.ac.ir; 4College of Computer Science and Technology, Changchun Normal University, Changchun 130032, China; 5School of Computing and Mathematical Sciences, University of Leicester, Leicester LE1 7RH, UK; shuihuawang@ieee.org; 6Department of Biological Sciences, Xi’an Jiaotong-Liverpool University, Suzhou 215123, China; 7School of Computer Science and Technology, Henan Polytechnic University, Jiaozuo 454000, China; 8Department of Information Technology, Faculty of Computing and Information Technology, King Abdulaziz University, Jeddah 21589, Saudi Arabia

**Keywords:** face image, multi-threshold segmentation, meta-heuristic optimization, Kapur’s entropy

## Abstract

Image segmentation methods have received widespread attention in face image recognition, which can divide each pixel in the image into different regions and effectively distinguish the face region from the background for further recognition. Threshold segmentation, a common image segmentation method, suffers from the problem that the computational complexity shows exponential growth with the increase in the segmentation threshold level. Therefore, in order to improve the segmentation quality and obtain the segmentation thresholds more efficiently, a multi-threshold image segmentation framework based on a meta-heuristic optimization technique combined with Kapur’s entropy is proposed in this study. A meta-heuristic optimization method based on an improved grey wolf optimizer variant is proposed to optimize the 2D Kapur’s entropy of the greyscale and nonlocal mean 2D histograms generated by image computation. In order to verify the advancement of the method, experiments compared with the state-of-the-art method on IEEE CEC2020 and face image segmentation public dataset were conducted in this paper. The proposed method has achieved better results than other methods in various tests at 18 thresholds with an average feature similarity of 0.8792, an average structural similarity of 0.8532, and an average peak signal-to-noise ratio of 24.9 dB. It can be used as an effective tool for face segmentation.

## 1. Introduction

Face-based research has received increasing attention as one of the most common and non-contact collection important biometric features. One of the hot topics in face image processing technology is object recognition [1,2,3]. Face segmentation and face recognition technologies have been used in various places, such as banks [4], schools [5], and libraries [6]. However, face recognition, person detection, and image processing techniques often depend on the effect of image segmentation. Rangayya et al. [7] used kernelized total Bregman divergence-based K-Means clustering-based segmentation technique in the proposed face recognition method to reduce noise interference on the segmentation effect, effectively improving face recognition. Khan et al. [8] developed an automatic facial image segmentation model based on conditional random fields to improve classification accuracy. Segundo et al. [9] embedded a segmentation technique based on edge detection, region clustering, and shape analysis into a face detection system to improve face recognition performance. Zhang et al. [10] proposed a multistep iterative segmentation algorithm to achieve fine segmentation of obscured characters and improve recognition accuracy. Efficient and accurate image segmentation techniques can help improve the performance of face recognition systems [11,12,13].

At present, there are many image segmentation methods, such as multi-threshold [14], region growth [15], edge detection [16], and deep learning [17,18]. At the same time, unsupervised learning-based image segmentation methods are currently available for unlabeled training data. For example, Xia et al. [19] proposed a new unsupervised learning segmentation network by combining two fully convolutional networks into one autoencoder inspired by the idea of semantic segmentation. Kim et al. [20] designed an end-to-end unsupervised image segmentation network consisting of argmax functions for normalization and differentiable clustering. Although the segmentation methods based on unsupervised learning do not need to train the neural network in advance, and can directly and unsupervised segmentation of a single image, not only greatly save the computational resources, and segmentation of significant targets more accurate features and advantages. However, such methods are not stable enough to perform the image segmentation task and cannot effectively extract texture features to segment the image into regions of overall significance when the same target has significant color differences. There are defects of confusing foreground and background in some images, over-reliance on color information, and little consideration of target spatial feature information. There is also the problem of poor robustness in unsupervised learning. It is difficult for such methods to place restrictions on neural networks speculatively outputting a sheet of results containing only one category, a design that is more prone to overfitting. Threshold-based image segmentation methods do not require a priori knowledge; the robust segmentation effect is excellent and is an efficient means of image segmentation. On the other hand, the typical exhaustive method for determining the best threshold will increase computing complexity while decreasing computational efficiency. Therefore, using a meta-heuristic optimization algorithm to search for the optimal threshold has become effective. Li et al. [21] proposed a threshold segmentation technique based on particle swarm optimization. Liu et al. [22] used the firework algorithm to find the optimal threshold set. Li et al. [23] used the biogeographic optimization algorithm to enhance multi-threshold image segmentation. Dutta et al. [24] proposed a multi-level image thresholding method based on the quantum genetic algorithm. The threshold segmentation method based on meta-heuristic optimization can obtain the optimal set of thresholds for segmentation more efficiently, which is considered a promising threshold segmentation method.

In recent times, optimization methods have experienced a surge in prominence, capturing the sustained interest of the community in swarm-based optimization, distributed optimization [25], robust optimization [26], multi-objective optimization [27], many objective cases [28], fuzzy optimization [29], etc. The optimization methods can be classified into two fundamental classes: deterministic and approximative techniques, rendering them amenable to addressing a wide spectrum of problem scenarios [30,31]. The meta-heuristic approaches stand as pivotal categories of optimization techniques deeply grounded in concepts such as mutation, crossover, and various iterative procedures. These methodologies enable the exploration of solution spaces independently of gradient information, provided that the newly generated solutions adhere to prescribed optimality criteria. Two of the most well-known approaches are genetic algorithms (GAs), which are based on the selection of fittest and survival values in nature [32,33]. However, these classes of swarm-based methods are prone to several risks, including weak mathematical models, low robustness issues, immature convergence, and stagnation possibility [34,35]. The meta-heuristic optimization algorithms can be utilized to solve the optimal solution of complex problems, such as image segmentation [36], feature selection [37], real-world optimization problems [38], bankruptcy prediction [39], scheduling optimization [40], multi-objective optimization [41], global optimization [42,43], target tracking [44], economic emission dispatch [45], feed-forward neural networks [46], and numerical optimization [47,48,49]. It has become one of the most popular optimization methods due to its excellent optimization ability. Common optimization algorithms include particle swarm optimization (PSO) [50], sine and cosine optimization algorithm (SCA) [51], whale optimization algorithm (WOA) [52], slime mould algorithm (SMA) [53,54], hunger games search (HGS) [55], Harris hawks optimization (HHO) [56], colony predation algorithm (CPA) [57], rime optimization algorithm (RIME) [58], the weighted mean of vectors (INFO) [59], Runge Kutta optimizer (RUN) [60], grey wolf optimization algorithm (GWO) [61], and other optimization methods. As an example, the development of a method to evaluate task offloading strategies within the context of Mobile Edge Computing (MEC) was facilitated by the utilization of the Sine and Cosine Algorithm (SCA) [62].

According to the No Free Lunch theorems for optimization [63], it is known that no one optimization method can perform well on all problems. Therefore, more and more improved optimization algorithms based on optimization strategies have been proposed to address the shortcomings of different optimization algorithms in the process of searching for the global optimal solution. To adapt the algorithm to the more complex problems to be optimized, the improvement of the original algorithm has become another research hotspot. For example, Yang et al. [64] utilized roundup search, the elite Lévy-mutation, and the decentralized foraging optimization techniques to enhance the performance of differential evolution for multi-threshold image segmentation. Zhang et al. [65] proposed an adaptive differential evolution with an optional external archive (JADE). Guo et al. [66] proposed a self-optimization approach for L-SHADE (SPS_L_SHADE_EIG). Qu et al. [67] proposed a modified sine cosine algorithm based on a neighborhood search and a greedy levy mutation (MSCA). Han and Li [68] proposed an improved genetic algorithm based on adaptive crossover probability and adaptive mutation probability.

PSO is an excellent optimization algorithm; with few parameters and easy implementation. The PSO and its variants have received extensive attention from researchers. Among many PSO variants, GWO not only inherits the advantages of PSO but also is an attempt to improve global optimization ability and convergence [69]. Therefore, improved GWOs based on optimization strategies have been proposed recently. Cai et al. [70] optimized a kernel extreme learning machine with an enhanced GWO. Choubey et al. [71] optimized the parameters of the multi-machine power system stabilizer. Li et al. [72] monitored the robot path through an enhanced GWO optimization area. Mehmood et al. [73] applied an improved grey wolf optimizer technique based on chaotic mapping to optimize problems such as autoregressive exogenous structural parameter optimization.

In thresholding optimization, local optimal or suboptimal solutions mistakenly considered the best set of thresholds can lead to incorrect segmentation. Thus, the critical information about the target is lost, resulting in the degradation of image segmentation quality. To obtain a method with high optimization accuracy and a high ability to jump out of the local optimum, this study proposed a GWO improvement method based on the cosmic wormhole strategy, denoted as WGWO. To distinguish background and object more efficiently and to consider the spatial information of pixels, Kapur’s entropy was used as the objective function of WGWO and combined with a nonlocal mean two-dimensional histogram to achieve high-quality segmentation of face images. From the Berkeley dataset [74] and Flickr-Faces-High-Quality dataset (FFHQ) [75], eight face images were selected for comparative experiments, and then the segmentation effects were verified by three image evaluation metrics. The experimental results show that the WGWO multi-threshold segmentation method achieves satisfactory segmentation results. The main contributions of this paper are as follows:

A multi-threshold image segmentation method based on optimization technique and 2D histogram is proposed, which is used to segment face images;An enhanced grey wolf optimizer based on cosmic wormhole strategy is proposed that is used to obtain the optimal segmentation threshold for the image.

The remainder of the paper is organized as follows. Section 2 describes the proposed WGWO. Section 3 presents the WGWO-based image segmentation method. In Section 4, the segmentation results have been verified and discussed. Section 5 summarizes the current work and explains the next research directions.

## 2. The Proposed WGWO

To improve the search efficiency of the optimal threshold set, this section details an improved grey wolf optimizer for image segmentation called WGWO.

### 2.1. Original GWO (a Variant of PSO)

As a PSO algorithm variant [76], GWO mainly consists of $S_{1}^{t}$, $S_{2}^{t}$ and $S_{3}^{t}$ to update the positions of the other particles in the population at the *t*-th iteration. $S_{1}^{t}$, $S_{2}^{t}$ and $S_{3}^{t}$ are biased by biasing the positions of individual particles in the population based on the top three optimal individuals in the population, $g_{1}^{t}$, $g_{2}^{t}$ and $g_{3}^{t}$, respectively, as shown in Equations (1)–(3).
(1)$$S_{1}^{t}=g_{1}^{t}-\left(2\varphi^{t}-1\right)\cdot r_{1}^{t}\cdot \left|2q_{1}^{t}\cdot g_{1}^{t}-X_{i}^{t}\right|$$
(2)$$S_{2}^{t}=g_{2}^{t}-\left(2\varphi^{t}-1\right)\cdot r_{2}^{t}\cdot \left|2q_{2}^{t}\cdot g_{2}^{t}-X_{i}^{t}\right|$$
(3)$$S_{3}^{t}=g_{3}^{t}-\left(2\varphi^{t}-1\right)\cdot r_{3}^{t}\cdot \left|2q_{3}^{t}\cdot g_{3}^{t}-X_{i}^{t}\right|$$
where $r_{1}^{t}$, $r_{2}^{t}$, $r_{3}^{t}$, $q_{1}^{t}$, $q_{2}^{t}$, $q_{3}^{t}$ all denote random numbers, obeying a random distribution between 0 and 1. $\varphi^{t}$ denotes an acceleration weighting factor decreasing from 2 to 0, $\varphi^{t}=2(1-t/T)$. $X_{i}^{t}$ denotes the position vector of the *i*-th particle at the *t*-th iteration.

Combining the position information of the three intermediates $S_{1}^{t}$, $S_{2}^{t}$, and $S_{3}^{t}$ are used to update the position of the *i*-th particle according to the rule of weighted averaging, as shown in Equation (4).
(4)$$X_{i}^{t+1}=\left(S_{1}^{t}+S_{2\;}^{t}+S_{3}^{t}\right)/3$$

### 2.2. Improved GWO (WGWO)

GWO has shown excellent optimization performance in various fields, e.g., power load forecast [77], power load forecast [78], etc. However, GWO still has insufficient segmentation accuracy for the image segmentation involved in this study. GWO is optimized in this study by combining the cosmic wormhole technique with population variety to improve the ability to avoid falling into the local optimum. Equations (8)–(11) show the mathematical model of the cosmic wormhole strategy [79].
(5)$$X_{i}^{t+1}=\left\{\begin{array}{c} \left\{\begin{array}{c} X_{i}^{t}+weight,r_{5}^{t}<0.5\\ X_{i}^{t}-weight,r_{5}^{t}\geq 0.5 \end{array}\right.,\;Ada<r_{4}^{t}\\ X_{i}^{t}\;,Ada\geq r_{4}^{t} \end{array}\right.$$
(6)$$\mathrm{weight}=M\cdot \left((\mathrm{UB}-\mathrm{LB})\cdot r_{6}^{t}+\mathrm{LB}\right)$$
(7)Ada= (T+4t)/5T
(8)$$M=1-t^{1/C}/T^{1/C}$$

In Equations (5)–(8), *Ada* is a probability parameter, *Ada* = [0.2, 1]. The candidate solution is decided whether to update by probability parameter *Ada*, *M* is a weight parameter, which controls the influence of random search on the current candidate solution through different search epochs. *C* is a constant of value 6. $r_{4}^{t}$, $r_{5}^{t}$, $r_{6}^{t}$ represent random numbers between 0 and 1.

To obtain a better solution, we proposed the WGWO method by introducing the cosmic wormhole strategy after the wolf population update, and the flowchart of WGWO is shown in Figure 1 (code has been publicly available at https://github.com/Forproject1111/WGWO, accessed on 8 October 2023). To calculate the time complexity of WGWO, we need to analyze the maximum number of iterations of the algorithm (*T*), the size of the population (*N*), and the variable size of the individuals (*D*). WGWO consists of population initialization, searching for prey, and cosmic wormhole strategy. Therefore, the time complexity of WGWO is O(((2*N* + 1) × *D*) × *T*).

## 3. Multi-Threshold Image Segmentation Method

### 3.1. The Basic Theory of Multi-Threshold Image Segmentation

The multi-threshold image segmentation method based on WGWO, NML 2-D histogram, and Kapur’s entropy reduces noise interference and improves segmentation efficiency.

#### 3.1.1. NML 2-D Histogram

The NML 2-D histogram [13] is composed of grayscale values and nonlocal means in a digital image, which reflects the grayscale size of pixels and information related to the pixel and neighborhood space. Assume that an image *I* of size *M* × *N* exists. *p* and *q* represent pixel points, and *X*() represents the pixel value. NML mathematical model of *I* is shown in Equations (9)–(12):(9)$$O(p)=\frac{\sum_{q\in \;I}X(q)\cdot \omega (p,q)}{\sum_{q\in \;I}\omega (p,q)}$$
(10)$$\omega (pq)=\exp^{-\frac{{\left|\mu (q\right)-\mu (p)|}^{2}}{\sigma^{2}}}$$
(11)$$\mu (p)=\frac{1}{n\times n}\sum_{i\in S(p)}I(i)$$
(12)$$\mu (q)=\frac{1}{n\times n}\sum_{i\in S(q)}I(i)$$
where *ω*(*p*, *q*) is a Gaussian weighting function. *σ* denotes the standard deviation. *μ*(*p*), *μ*(*q*) represent the local mean of *p*, *q* pixels. *x* is a pixel in the image *I*, and *S*(*x*) is a filter matrix of size *n* × *n* around the pixel *x*, *x* ∈ [1, *M*], *y* ∈ [1, *N*].

For each pixel, *I*(*x*, *y*) in *I*, *x*∈[1, *M*], *y*∈[1, *N*], the corresponding grayscale *f*(*x*, *y*) and the nonlocal mean *g*(*x*, *y*) can be calculated. Then, *i* and *j* are used to denote *f*(*x*, *y*) and *g*(*x*, *y*), respectively. *h*(*i*, *j*) denotes the vertical coordinate of the two-dimensional histogram, i.e., the number of occurrences of the gray-NML pair. Finally, *h*(*i*, *j*) is normalized to obtain $P_{\mathrm{ij}}$ to construct a greyscale nonlocal mean two-dimensional histogram, as shown in Figure 2.

#### 3.1.2. Kapur’s Entropy

To ensure that the segmented image obtained the maximum amount of information between the background and the target, the concept of Kapur’s entropy [13,80] was introduced in this study. Kapur’s entropy was used as a physical quantity to measure the amount of information distributed over the target and background regions. The greater Kapur’s entropy, the better the image segmentation quality. The following describes the process for segmenting images with several thresholds using Kapur’s entropy. The objective function is expressed as computing the entropy of *L* − 1 image segments and summing them. The objective function *F* expression of Kapur’s entropy is shown in Equations (13)–(15).
(13)$$\varphi (s,t)=H_{1}+H_{2}+\ldots +H_{L-1}$$
(14)$$\left\{\begin{array}{c} H_{1}=-\sum_{i=0}^{s_{1}}\;\sum_{j=0}^{t_{1}}\;\frac{P_{\mathrm{ij}}}{P_{1}}ln\frac{P_{\mathrm{ij}}}{P_{1}}\\ H_{2}=-\sum_{i=t_{1}+1}^{s_{2}}\;\sum_{j=t_{1}+1}^{t_{2}}\;\frac{P_{\mathrm{ij}}}{P_{2}}ln\frac{P_{\mathrm{ij}}}{P_{2}}\\ H_{L-1}=-\sum_{i=s_{L-2}+1}^{s_{L-1}}\sum_{j=t_{L-2}+1}^{t_{L-1}}\frac{P_{\mathrm{ij}}}{P_{L-1}}\ln\frac{P_{\mathrm{ij}}}{P_{L-1}} \end{array}\right.$$
(15)$$\left\{\begin{array}{c} P_{1}=\sum_{i=0}^{s_{1}}\sum_{j=0}^{t_{1}}P_{\mathrm{ij}}\\ P_{2}=\sum_{i=t_{1}+1}^{s_{2}}\sum_{j=t_{1}+1}^{t_{2}}P_{\mathrm{ij}}\\ P_{L-1}=\sum_{i=s_{L-2}+1}^{s_{L-1}}\sum_{j=t_{L-2}+1}^{t_{L-1}}P_{\mathrm{ij}} \end{array}\right.$$
where $H_{i}$ represents the entropy of the *i*-th image segment. $\left\{t_{1},t_{2}\ldots ,L-1\right\}$ represents the grayscale value of the grayscale image and $\left\{s_{1},s_{2}\ldots ,L-1\right\}$ characterizes the grayscale value of the nonlocal mean image.

### 3.2. Image Segmentation Method

The flowchart of segmentation based on WGWO, Kapur’s entropy, and NML 2-D histogram are shown in Figure 3. As shown in Figure 3, the input image is converted into a grayscale image and a nonlocal mean filtered image. Then a 2D histogram is calculated using the grayscale information of the grayscale image and the nonlocal mean filtered image. The Kapur’s entropy of the two-dimensional histogram is used as an objective function to optimize the segmentation threshold of the image using the proposed WGWO algorithm, which ultimately segments the image into multiple regions. The pseudo-code for segmentation is shown in Algorithm 1.
**Algorithm 1** The flow of image segmentation methodStep 1: Input digital image *I*, which has a size of *M* × *N*. The grayscale image *F* is obtained by graying out the image *I*;Step 2: The grayscale image F is nonlocal mean filtered to obtain the nonlocal mean image G according to Equations (9)–(12);Step 3: A two-dimensional image histogram is constructed using the grayscale values and nonlocal means in F and G;Step 4: Compute the two-dimensional Kapur’s entropy according to Equations (13)–(15);Step 5: Kapur’s entropy of the two-dimensional histogram is optimized using WGWO;Step 6: Multi-threshold image segmentation is performed according to the optimal threshold set to obtain pseudo-color and gray images.

## 4. Experiment Simulation and Analysis

In this section, the performance of the WGWO-based multi-threshold segmentation method proposed in this paper is verified. In addition, all programs were run using Matlab 2018b on a Windows 10 OS-based computer with an Intel CPU i5-11400H (2.70 GHz) and 16 GB of RAM.

### 4.1. IEEE CEC2020 Benchmark Dataset Experiment

In this subsection, this study conducted an ablation experiment and parameter experiment based on IEEE CEC2020 [81] test components to present the global optimization capability of WGWO. The details of IEEE CEC2020 are shown in Table 1. In addition, to ensure the fairness of the experiments, the public parameters of all test methods were set uniformly, the maximum number of function evaluations was set to 300,000, the population size was set to 30, the dimensionality was set to 30, and the number of independent runs of the program was set to 30.

First, the ablation experiment was introduced in this subsection to justify the improved strategy. Where WGWO1 used only $X_{i}^{t+1}=X_{i}^{t}+\mathrm{weight}$ as the update formula in Equation (5). WGWO2 used only $X_{i}^{t+1}=X_{i}^{t}-\mathrm{weight}$ as the updated formula. Table 2 shows the ranking of the ten benchmark functions and the final ranking in the ablation experiment. Where the optimal results were bolded. From the table, it can be seen that WGWO and WGWO2 have the best results in four cases each. However, the average ranking of 1.90 for WGWO was the best among the four methods. Therefore, WGWO was selected as the threshold optimization method for image segmentation in this study.

Second, the values of the parameters are one of the reasons that affect the performance of the algorithm. Among them, *Ada* and *C* are two key parameters in WGWO. Therefore, this study focused on testing the parameter sensitivity of *Ada* and *C*. Table 3 and Table 4 show the ranking results and the final ranking of *Ada* and *C*, respectively. As can be seen in Table 3, the combined ability was significantly better than that of the other versions of WGWO despite the relatively poor results for values in the range [0.3, 1] on F1 and F5. Furthermore, the final ranking in Table 4 shows that WGWO was optimized the most when the C value was 6. Based on the results of the sensitivity tests of the two parameters, we fine-tuned *Ada* and *C*. A randomized value range of [0.3, 1] for *Ada* and a value of 6 for *C* were finally determined as the parameter values for the final version of WGWO for subsequent experiments.

In conclusion, based on the results of the ablation experiment and the parameter sensitivity experiment described above, it can be concluded that the best version of WGWO was applied to optimize the threshold for multi-threshold image segmentation.

### 4.2. Multi-Threshold Face Image Segmentation Experiment

To provide quality face recognition data, this subsection reports the testing of image segmentation performance of the proposed method.

#### 4.2.1. Experimental Settings

Test images of faces from the Berkeley dataset and the Flickr-Faces-High-Quality dataset were selected as validation materials, as shown in Figure 4. The size of the images was 321 × 481, where images A to D were from the Berkeley dataset and images E to H were from the FFHQ dataset. Threshold values were set in the range of 0 to 255. The maximum number of iterations was set to 100. The population size was set to 30, and the number of independent runs was set to 30.

WGWO was compared to GWO [61], PSO [50], WOA [52], BLPSO [82], IGWO [70], HLDDE [64], SCADE [83], and IWOA [84] in multi-threshold image segmentation experiments. The initialization parameters of the nine algorithms are shown in Table 5. To verify the segmentation effect of WGWO, this paper examined the segmentation effect through three evaluation methods: feature similarity (FSIM) [85], structural similarity (SSIM) [86], and peak signal-to-noise ratio (PSNR) [87]. The details of these indicators are shown in Table 6. The segmentation results were analyzed for the significance of differences using the Wilcoxon signed-rank test (WSRT) [88]. In WSRT, if the *p*-value is less than 0.05 and WGWO is superior to the comparison method, the advantage of WGWO performance is statistically significant and is denoted by ‘+’. If the *p*-value is less than 0.05 and WGWO is inferior to the comparison method, the advantage of the comparison method performance is statistically significant and is denoted by ‘−’. If the *p*-value is greater than or equal to 0.05, the performance of WGWO and the comparison method can be approximated as equal and is denoted by ‘=’.

#### 4.2.2. Image Segmentation Experiment

This experiment first demonstrates nine algorithms for segmenting visual images in the Berkeley and FFHQ datasets at a threshold level of 18. Figure 5, Figure A1, Figure A2 and Figure A3 show the results of pseudo-color segmented images and gray segmented images from image A to image H. It can be seen directly from the figures that SCADE and IWOA (in Figure 5, Figure A1, Figure A2 and Figure A3) had poor segmentation effects, and HLDDE (in Figure A2 and Figure A3) also had a poor segmentation effect on image F and image H. Subsequently, Figure 6 shows that WGWO had excellent segmentation results for all eight images at different thresholds. In this case, important information was lost in the segmented images at lower thresholds, while the high threshold segmentation retained more image detail. It is important to note that the visual result only shows the segmentation effect. The experimental results based on FSIM, PSNR, and SSIM reflect the quality of the segmented image more objectively. As a result, the FSIM, PSNR, and SSIM results of WGWO were further analyzed and discussed.

Table 7 demonstrates the comparison results of FSIM between the proposed method and other methods at 4 threshold levels. From all 4 threshold levels, WGWO had the best average rankings among the compared algorithms, and even though the results of WGWO were weaker than those of GWO at 3 images in the segmentation results at 5-level and 8-level thresholds, WGWO was better as a whole. It indicates that the segmented images obtained based on the WGWO method are better able to portray the local features of the target, as well as being more in line with the human visual system’s perception of low-level features. Table 8 demonstrates the PSNR comparison results of each segmentation method. A comprehensive analysis of the data in Table 8 shows that the WGWO-based segmentation method was also excellent and stable, indicating that the WGWO-based segmentation method has less distortion and higher image quality in the segmentation process compared with the original image. Table 9 demonstrates the SSIM results of these compared methods. As can be seen from Table 9, the proposed WGWO-based segmentation method dominated, which indicates that the proposed method has less distortion and is more in line with the requirements of the human visual system. It is worth noting that as a variant of PSO, GWOs were more suitable for solving the optimization problem of segmentation thresholds compared to the PSO baseline method (in Table 7, Table 8 and Table 9). In conclusion, the segmentation performance of WGWO has been validated by three image segmentation quality assessment metrics at multiple threshold levels.

In addition to exploring the image segmentation performance of WGWO in a conventional setting, this study further explored the stability of the proposed method when dealing with a larger number of population agents dealing with high threshold segmentation problems. In this study, we set the initial population size to 100 and the threshold segmentation level to 18. Table 10 shows the ranking of the three segmentation metrics and the comparison results based on WSRT. The table shows that the multi-threshold image segmentation method of WGWO is still the best performer with stable image segmentation performance under the condition of a larger population size.

In this study, time cost and convergence of threshold optimization methods are one of the evaluation methods of the algorithm’s performance, especially the convergence based on Kapur entropy, which is the key for the algorithm to obtain the optimal segmentation threshold. Figure 7 shows the average time cost of 30 experiments of 9 algorithms on different threshold levels for all the images processed. The observation shows that the proposed method ranked third in terms of time cost on each threshold. Moreover, the time cost of each algorithm grew as the threshold level increased. Figure 8 shows the convergence curves for all methods. Figure 8 depicts the convergence curves of nine comparison techniques for Kapur’s entropy optimization on eight images. The optimization of Kapur’s entropy can be considered a maximum optimization problem, so a higher entropy value means that more useful information is retained, making the segmentation better. The following points can be seen by observing the convergence curves of the optimization of eight images. First, a simple examination of WGWO’s eight convergence curves reveals that, although its convergence speed was not the fastest in the whole convergence process, its convergence accuracy was superior to that of other algorithms. It is worth noting that WGWO has a strong ability to prevent premature convergence. Second, the convergence curve of the WOA algorithm in the early and middle iterations was above all algorithms. The slope of the convergence curve of the WGWO algorithm became larger around the 90th iteration, which resulted in the best fitness of WGWO among all methods. Third, compared to PSO, GWO was more suitable to deal with the image segmentation problem based on Kapur’s entropy, and GWO had higher convergence accuracy and a better ability to jump out of the local optimal solution. Finally, by comparing the convergence curves of WGWO and GWO, it can be seen that the changes in the two convergence curves were very similar, but the convergence accuracy of WGWO was better than that of GWO, which shows that the introduction strategy enhances the optimization ability of the algorithm.

In conclusion, WGWO has the best optimization ability compared to the other five algorithms and can complete higher-quality multi-threshold image segmentation. Of course, it also can be applied to many other fields, such as machine learning models [89], image denoising [90], medical signals [91], structured sparsity optimization [92], renal pathology image segmentation [93], mental health prediction [94], lung cancer diagnosis [95], computer-aided medical diagnosis [96], MRI reconstruction [97], and power distribution network [98].

## 5. Conclusions

This study proposed a grey wolf optimization algorithm based on the cosmic wormhole strategy. The population position update mechanism was optimized to improve the convergence accuracy of the algorithm, which can help the algorithm jump out of the local optimum. A multi-threshold image segmentation method based on WGWO was then used to segment the face images. The facial picture is then segmented using a multi-threshold image segmentation approach based on WGWO. The experimental results show that WGWO makes obtaining a set of threshold values suitable for face image segmentation easier. The proposed method is also verified to have better segmentation performance than other methods by three image quality evaluation criteria. In conclusion, the proposed method can support intelligent library face recognition technology more effectively.

Although the WGWO-based image segmentation method proposed in this paper can provide better quality segmented images for face recognition systems, there are still some shortcomings. First, there is still potential to improve the optimization performance of WGWO. In addition, this paper does not explore the optimal threshold level. The above two points are still areas that authors need to further investigate. Additionally, it is intriguing to include parallel computing methods into the framework for multi-threshold picture segmentation to boost computational efficiency. 

## Figures and Tables

**Figure 1 biomimetics-08-00484-f001:**
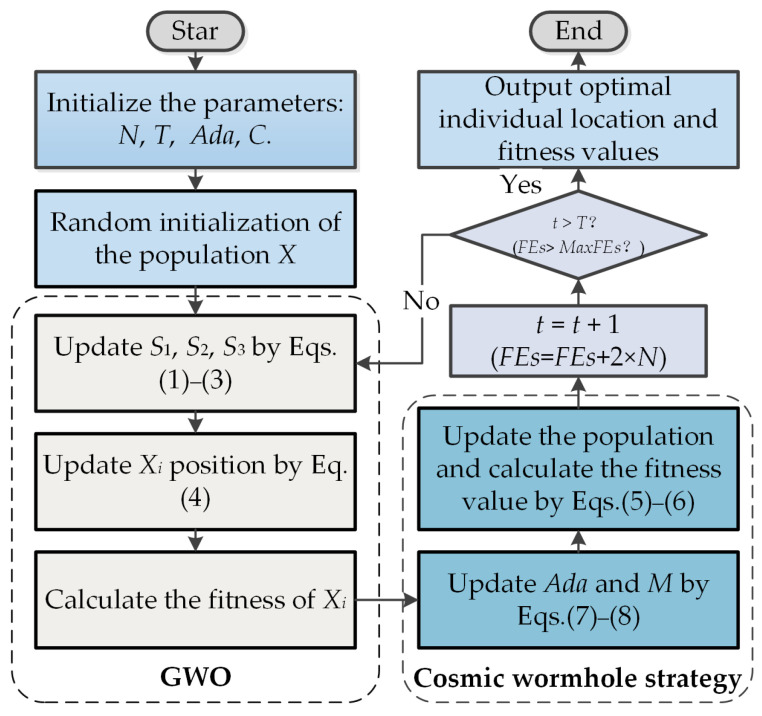
Flowchart of WGWO.

**Figure 2 biomimetics-08-00484-f002:**
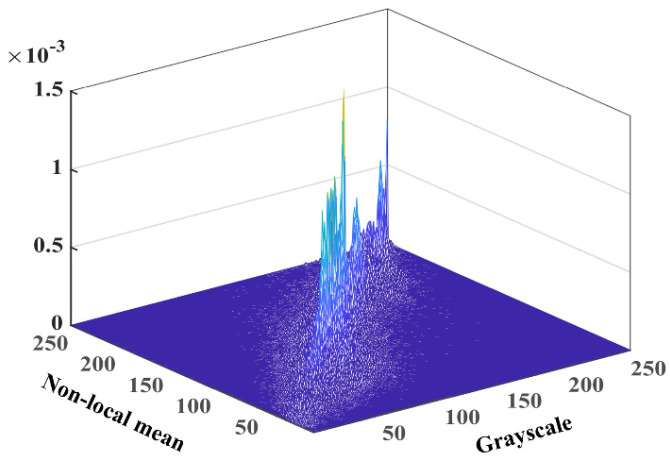
NML 2−D histogram.

**Figure 3 biomimetics-08-00484-f003:**
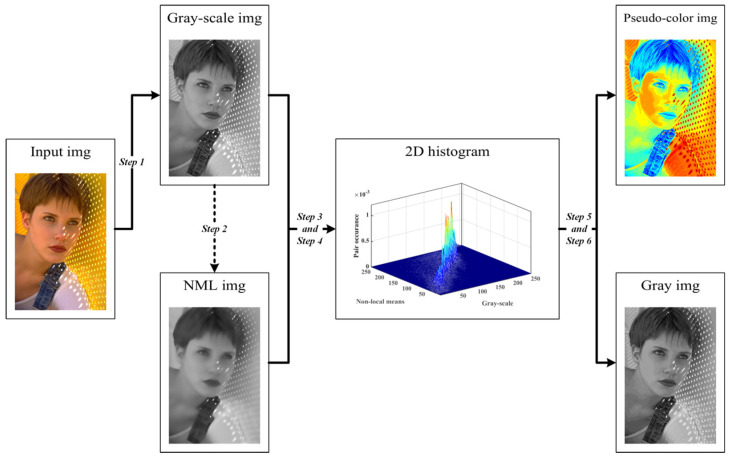
The flow chart of the multi−threshold image segmentation process.

**Figure 4 biomimetics-08-00484-f004:**
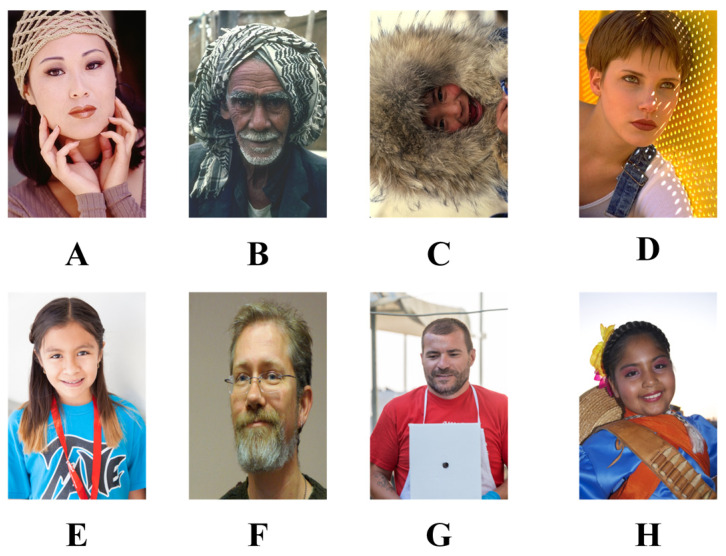
Face images from the image segmentation dataset.

**Figure 5 biomimetics-08-00484-f005:**
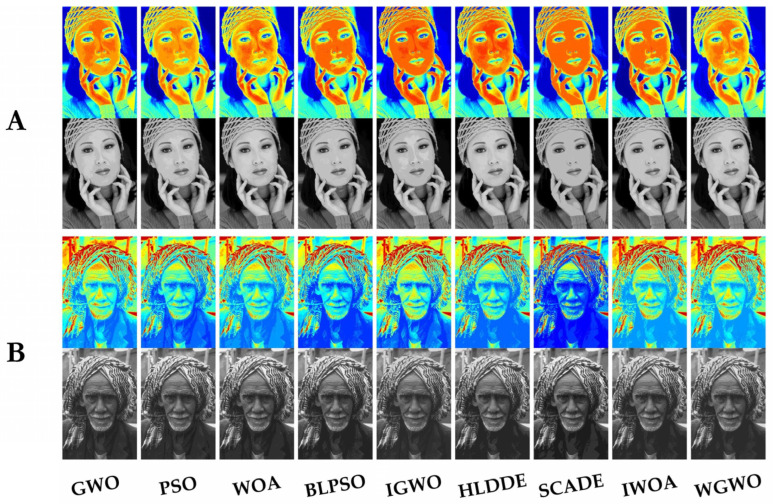
Segmentation results of images (**A**,**B**).

**Figure 6 biomimetics-08-00484-f006:**
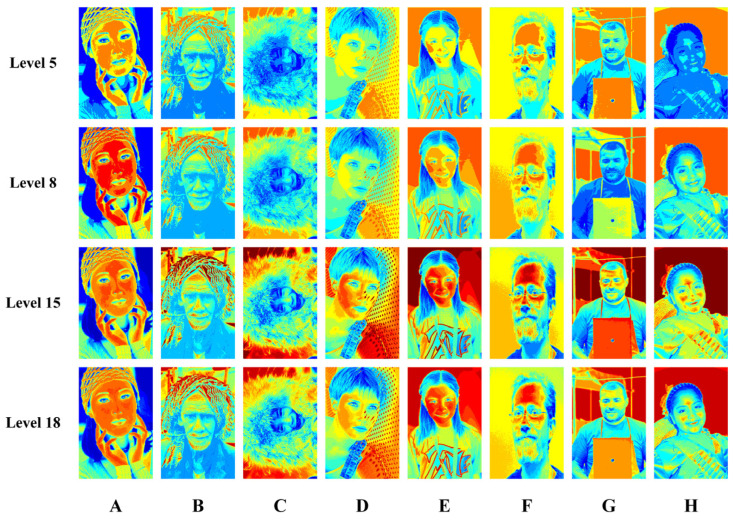
Segmentation results of WGWO.

**Figure 7 biomimetics-08-00484-f007:**
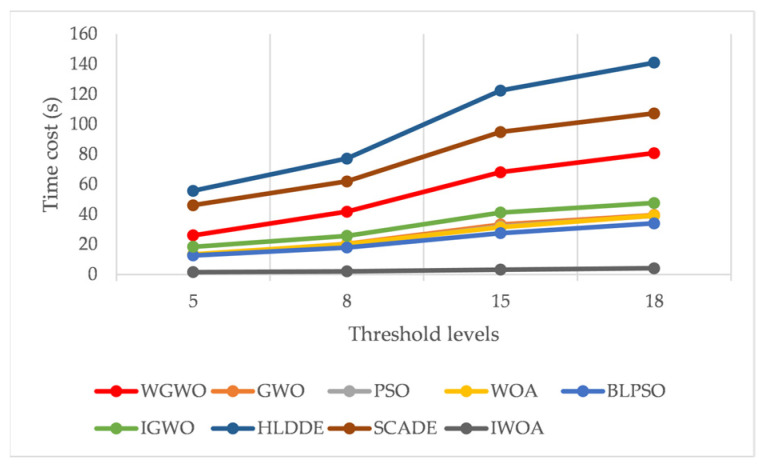
Time cost of each algorithm.

**Figure 8 biomimetics-08-00484-f008:**
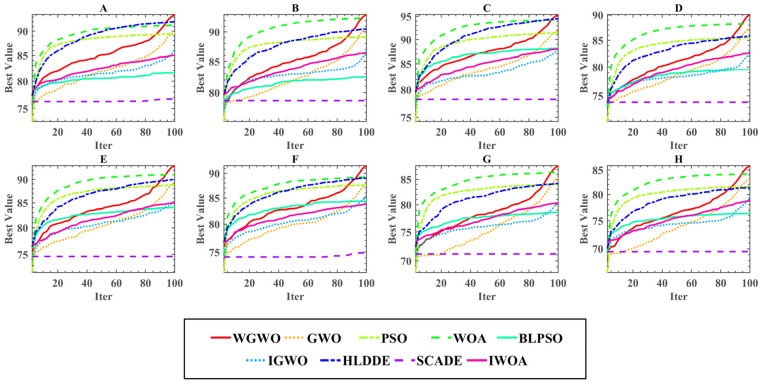
Convergence curves of each algorithm.

**Table 1 biomimetics-08-00484-t001:** Details of IEEE CEC2020 benchmark functions.

Class.	No.	Functions	Fmin
Unimodal Function	F1	Shifted and Rotated Bent Cigar Function	100
Basic Functions	F2	Shifted and Rotated Schwefel’s Function	1100
	F3	Shifted and Rotated Lunacek bi-Rastrigin Function	700
	F4	Expanded Rosenbrock’s plus Griewangk’s Function	1900
Hybrid Functions	F5	Hybrid Function 1 (N = 3)	1700
	F6	Hybrid Function 2 (N = 4)	1600
	F7	Hybrid Function 3 (N = 5)	2100
Composition Functions	F8	Composition Function 1 (N = 3)	2200
	F9	Composition Function 2 (N = 4)	2400
	F10	Composition Function 3 (N = 5)	2500
Search range: [–100, 100]*^D^*			

**Table 2 biomimetics-08-00484-t002:** Comparative results of the ablation experiment.

No.	WGWO	WGWO1	WGWO2	GWO
F1	2	**1**	3	4
F2	**1**	2	3	4
F3	2	3	**1**	4
F4	2	3	**1**	4
F5	**1**	2	3	4
F6	**1**	2	3	4
F7	**1**	3	2	4
F8	3	**1**	2	4
F9	3	2	**1**	4
F10	3	2	**1**	4
Result	**1 (1.90)**	3 (2.10)	2 (2.00)	4 (4.00)

**Table 3 biomimetics-08-00484-t003:** Sensitivity experiment of the parameter *Ada.*

No.	[0, 1]	[0.1, 1]	[0.2, 1]	[0.3, 1]	[0.4, 1]	[0.5, 1]	[0.6, 1]	[0.7, 1]	[0.8, 1]	[0.9, 1]
F1	6	8	4	10	5	3	**1**	9	2	7
F2	7	8	10	6	4	2	5	**1**	9	3
F3	9	4	**1**	2	3	5	6	7	8	10
F4	**1**	2	4	3	5	6	9	7	10	8
F5	**1**	2	6	8	5	9	3	4	7	10
F6	2	4	9	6	7	**1**	5	3	10	8
F7	5	**1**	2	3	10	7	4	6	9	8
F8	9	5	**1**	2	4	3	8	6	7	10
F9	10	8	2	**1**	3	4	6	5	7	9
F10	**1**	4	6	3	9	5	8	10	2	7
Result	5 (5.1)	4 (4.6)	2 (4.5)	**1 (4.4)**	6 (5.5)	2 (4.5)	6 (5.5)	8 (5.8)	9 (7.1)	10 (8)

**Table 4 biomimetics-08-00484-t004:** Sensitivity experiment of the parameter *C.*

No.	*C* (1)	*C* (2)	*C* (3)	*C* (4)	*C* (5)	*C* (6)	*C* (7)	*C* (8)	*C* (9)
F1	9	8	7	6	5	3	2	4	**1**
F2	9	8	7	5	6	3	**1**	4	2
F3	8	9	7	6	5	2	4	**1**	3
F4	7	9	8	2	3	**1**	6	4	5
F5	9	8	7	4	6	**1**	3	2	5
F6	9	4	3	7	**1**	2	6	8	5
F7	9	7	8	4	5	6	2	3	**1**
F8	6	5	3	8	9	4	7	**1**	2
F9	5	**1**	2	4	3	7	6	8	9
F10	9	8	7	**1**	3	2	4	6	5
Result	9 (8)	8 (6.7)	7 (5.9)	6 (4.7)	5 (4.6)	**1 (3.1)**	3 (4.1)	3 (4.1)	2 (3.8)

**Table 5 biomimetics-08-00484-t005:** Nine algorithm parameter settings.

Methods	Parameters	Criteria
WGWO	φ∈[2 0]	Original paper [61]
	*Ada* ∈[0.3 1]	Parameter sensitivity analysis (Section 4.1)
	*C* = 6	Parameter sensitivity analysis (Section 4.1)
GWO	*a* ∈[2 0]	Original paper [61]
PSO	ω *=* $1,\;c_{1}$ $=2,\;c_{2}$ = 2	Original paper [50]
WOA	$a_{1}\in$$[2\;0],\;a_{2}\in$[−1 −2], *b* = 1	Original paper [52]
BLPSO	*c* =1.496, I=1, E=1, ω ∈ [0.9 0.2]	Original paper [82]
IGWO	$\beta_{\mathrm{num}}=\;10;\;\Omega_{\mathrm{num}}=\;15$	Original paper [70]
HLDDE	J∈[0 2], DR∈[0 0.4]	Original paper [64]
SCADE	*a* $=\;2,\;P_{c}$ = 0.8	Original paper [83]
IWOA	$a_{1}\in$$[2\;0],\;a_{2}\in$[−1 −2]	Original paper [84]

**Table 6 biomimetics-08-00484-t006:** Details of the three image evaluation metrics.

Metrics	Formulas	Remarks
FSIM [85]	$\mathrm{FSIM}=\frac{\sum_{x\in \Omega }S_{L}(x)\cdot {\mathrm{PC}}_{m}(x)}{\sum_{x\in \Omega }{\mathrm{PC}}_{m}(x)}$	FSIM is an image quality assessment method based on phase consistency features and gradient features complementing each other.
SSIM [86]	$\mathrm{SSIM}=\frac{(2\mu_{I}\mu_{K}+C_{1})(2\sigma_{\mathrm{IK}}-C_{2})}{\left(\mu_{I}^{2}+\mu_{K}^{2}+C_{1})(\sigma_{I}^{2}+\sigma_{K}^{2}+C_{2}\right)}$	SSIM is a similarity assessment based on the luminance, contrast, and structure of the original image and the segmented image, which is a full-reference image quality evaluation index more in line with human vision’s judgment of image quality.
PSNR [87]	$\mathrm{PSNR}=10{\cdot \log}_{10}{((\mathrm{peak}}^{2})/\mathrm{MSE})$	PSNR represents the ratio of the maximum possible power of a signal to the destructive noise power affecting its representation accuracy and is an objective full-reference image quality evaluation index.

**Table 7 biomimetics-08-00484-t007:** FSIM ranking of each algorithm at four thresholds.

Methods	5-Level			8-Level			15-Level			18-Level		
	+/−/=	Mean	Rank	+/−/=	Mean	Rank	+/−/=	Mean	Rank	+/−/=	Mean	Rank
WGWO	**~**	**1.63**	**1**	**~**	**1.38**	**1**	**~**	**1.63**	**1**	**~**	**1.38**	**1**
GWO	2/3/3	2.38	2	2/3/3	1.88	2	2/0/6	2	2	3/0/5	2.5	2
PSO	6/1/1	4	4	6/0/2	3.75	3	6/0/2	3.88	4	5/0/3	3.63	4
WOA	5/0/3	5.63	5	6/0/2	3.75	3	4/0/4	2.88	3	2/0/6	2.75	3
BLPSO	5/0/3	6.00	7	8/0/0	7.25	7	8/0/0	7.38	8	8/0/0	7.38	8
IGWO	5/0/3	3.88	3	8/0/0	5.13	5	8/0/0	6.88	7	8/0/0	6.88	7
HLDDE	8/0/0	5.63	5	8/0/0	5.38	6	8/0/0	5.00	5	8/0/0	5.13	5
SCADE	8/0/0	8.50	9	8/0/0	9.00	9	8/0/0	9.00	9	8/0/0	9.00	9
IWOA	8/0/0	7.38	8	8/0/0	7.50	8	8/0/0	6.38	6	8/0/0	6.38	6

**Table 8 biomimetics-08-00484-t008:** PSNR ranking of each algorithm at the four thresholds.

Methods	5-Level			8-Level			15-Level			18-Level		
	+/−/=	Mean	Rank	+/−/=	Mean	Rank	+/−/=	Mean	Rank	+/−/=	Mean	Rank
WGWO	**~**	**2.00**	**1**	**~**	**1.88**	**1**	**~**	**1.38**	**1**	**~**	**1.63**	**1**
GWO	3/2/3	2.25	2	1/2/5	2	2	1/1/6	2.13	2	0/0/8	2.38	2
PSO	6/1/1	3.63	3	7/0/1	4.38	4	7/0/1	4.88	5	7/0/1	5	5
WOA	5/0/3	4.88	5	6/0/2	4.00	3	3/0/5	2.88	3	2/0/6	2.88	3
BLPSO	4/0/4	5.25	6	7/1/0	6.00	7	7/0/1	6.63	6	7/0/1	6.00	6
IGWO	5/0/3	4.75	4	7/0/1	5.25	6	7/0/1	6.75	7	8/0/0	6.88	7
HLDDE	5/0/3	5.50	7	7/1/0	4.88	5	8/0/0	4.50	4	5/0/3	4.25	4
SCADE	8/0/0	9.00	9	7/0/1	9.00	9	8/0/0	9.00	9	8/0/0	8.88	9
IWOA	8/0/0	7.75	8	8/0/0	7.63	8	8/0/0	6.88	8	8/0/0	7.13	8

**Table 9 biomimetics-08-00484-t009:** SSIM ranking of each algorithm at the four thresholds.

Methods	5-Level			8-Level			15-Level			18-Level		
	+/−/=	Mean	Rank	+/−/=	Mean	Rank	+/−/=	Mean	Rank	+/−/=	Mean	Rank
WGWO	**~**	**2.50**	**1**	**~**	**1.63**	**1**	**~**	**1.50**	**1**	**~**	**1.25**	**1**
GWO	3/1/4	2.63	2	5/1/2	2.63	3	4/1/3	3.00	3	4/1/3	2.88	2
PSO	2/2/4	2.75	3	2/2/4	2.00	2	2/0/6	2.75	2	4/0/4	3.00	3
WOA	4/0/4	5.88	6	6/0/2	4.38	4	3/0/5	3.63	4	3/0/5	3.50	4
BLPSO	7/0/1	5.63	5	8/0/0	7.00	8	7/0/1	6.88	7	7/0/1	7.13	8
IGWO	4/0/4	3.63	4	7/0/1	4.75	5	8/0/0	6.88	7	8/0/0	7.00	7
HLDDE	8/0/0	6.50	7	8/0/0	6.88	6	7/0/1	5.00	5	7/0/1	5.25	5
SCADE	7/0/1	8.00	9	7/0/1	8.88	9	8/0/0	9.00	9	8/0/0	8.75	9
IWOA	6/0/2	7.50	8	7/0/1	6.88	6	8/0/0	6.38	6	8/0/0	6.25	6

**Table 10 biomimetics-08-00484-t010:** Segmentation stability ranking and WSRT comparison results in FSIM, PSNR, and SSIM.

Metrics	Items	GWO	PSO	WOA	BLPSO	IGWO	HLDDE	SCADE	IWOA	WGWO
FSIM	+/−/=	7/0/1	6/1/1	3/1/4	8/0/0	8/0/0	8/0/0	8/0/0	8/0/0	**~**
	Mean	3.75	3.00	1.75	7.63	5.88	5.38	8.88	7.13	**1.63**
	Rank	4	3	2	8	6	5	9	7	**1**
PSNR	+/−/=	6/0/2	1/1/6	3/1/4	7/0/1	8/0/0	6/0/2	8/0/0	8/0/0	**~**
	Mean	4.75	2.50	2.13	6.63	5.88	4.63	8.88	7.75	**1.88**
	Rank	5	3	2	7	6	4	9	8	**1**
SSIM	+/−/=	5/1/2	4/0/4	1/1/6	7/0/1	7/0/1	7/0/1	8/0/0	8/0/0	**~**
	Mean	2.88	3.50	2.63	7.50	5.38	5.38	8.75	7.38	**1.63**
	Rank	3	4	2	8	5	5	9	7	**1**

## Data Availability

The data is available at https://github.com/Forproject1111/WGWO, accessed on 8 October 2023.

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
