# Peer review of "Face Image Segmentation Using Boosted Grey Wolf Optimizer"

_biomimetics, 2023, doi:10.3390/biomimetics8060484_

Round 1
Reviewer 1 Report
In this work, a multi-threshold image segmentation framework is proposed. The proposed framework is based on meta-heuristic optimization method and Kapur's entropy. The meta-heuristic optimization method was optimized to optimize the Kapur's entropy of the image.
The manuscript is nice; however, the following comments need to be addressed carefully:
Abstract:
1 – The problem statement requires be more elaborated.
2 – The results in terms of improvement ratio between the proposed and existing works need to be included at the end of the abstract.
Introduction Section :
3 – The contribution list need to be revised such that the third point need to be eliminated
4 – More works need to be included in this section. For example: a) doi: 10.3390/s22239209, and b) doi: 10.3390/math10152721 .
The proposed WGWO Section :
5 – equations from other sources need to be referenced .
Multi-threshold image segmentation method Section :
6 - equations from other sources need to be referenced .
7 – Pseudo code algorithm for the algorithm need to be included to make it more clear to the readers .
8 – In the comparison, references for other algorithms need to be included in the figures and/o tables .
Conclusion Section :
9 – This section is fine. No Comments .
There are some typos and minor errors need to be fixed.
Author Response
[The comments of Reviewer #1]
Comment 1:
Abstract: The problem statement requires be more elaborated.
- Thanks for your valuable the reminder. Under your reminder, we have described the research questions in the abstract more clearly and explicitly. In addition, modified contents and added contents are marked in blue font in the new manuscript. With kind regards. (Lines 17 to 21)
Comment 2:
Abstract: The results in terms of improvement ratio between the proposed and existing works need to be included at the end of the abstract.
- Thanks for your valuable the reminder. Under your reminder, we have shown the quantitative results at the end of the abstract. In addition, modified contents and added contents are marked in blue font in the new manuscript. With kind regards. (Lines 29 to 31)
Comment 3:
Introduction Section: The contribution list need to be revised such that the third point need to be eliminated.
- Thanks for your valuable the reminder. Under your reminder, we have eliminated the third point from the list of contributions. With kind regards.
Comment 4:
Introduction Section: More works need to be included in this section. For example: a) doi: 10.3390/s22239209, and b) doi: 10.3390/math10152721.
- Thanks for your valuable the reminder. Under your reminder, we have introduced these references into this section. With kind regards.
Comment 5:
The proposed WGWO Section: Equations from other sources need to be referenced.
- Thanks for your valuable the reminder. Under your reminder, we have introduced references in related techniques in subsection 2.2. With kind regards.
Comment 6:
Multi-threshold image segmentation method Section: Equations from other sources need to be referenced.
- Thanks for your valuable the reminder. Under your reminder, we have introduced references in related techniques in subsection 3.1. With kind regards.
Comment 7:
Multi-threshold image segmentation method Section: Pseudo code algorithm for the algorithm need to be included to make it more clear to the readers.
- Thanks for your valuable the reminder. Under your reminder, we have re-described this section and added pseudo-code. In addition, modified contents and added contents are marked in blue font in the new manuscript. With kind regards. (Lines 176 to 185)
Comment 8:
Multi-threshold image segmentation method Section: In the comparison, references for other algorithms need to be included in the figures and/o tables.
- Thanks for your valuable the reminder. Under your reminder, we have introduced references to the comparison methods at the very beginning of our experimental section as well as in the table. In addition, modified contents and added contents are marked in blue font in the new manuscript. With kind regards. (Lines 235 to 236, Line 246)
Reviewer 2 Report
This paper presents a work on developing a multi-threshold face image segmentation using Cosmic Wormhole Strategy Enhanced Gray Wolf Optimization. The introduction part is good, where the authors have reviewed many recent related articles. The method has been described clearly, and the number of data used for evaluations are adequate. However, for further improvements:
1) From Figure 1, it is shown that the optimization process stops based on the maximum number of iterations allowed (T). Why not also include the stopping condition based on the fitness measures too?
2) When we do the benchmarks with other methods, better to provide the citations to those methods, including in the figures and tables. For example, cite the methods in Figure 5. By doing so, it is easier to see which one is the proposed method. Besides, usually the proposed method is located last. Thus, in my opinion, better to put WGWO to the right side of the figure.
3) In Table 6, it is not clear about which figures are being compared. For example, for SSIM, it compares the original image with the segmented image. Why not compare the segmented image with the ground truth? Similarly, for PSNR, how MSE is being calculated?
4) Are there the ground truth? If yes, better to include the ground truth in Figure 5, Figure A1, Figure A2 and Figure A3.
Author Response
[The comments of Reviewer #2]
Comment 1:
From Figure 1, it is shown that the optimization process stops based on the maximum number of iterations allowed (T). Why not also include the stopping condition based on the fitness measures too?
- I apologize for my carelessness. Under your reminder I have added a description of the number of function evaluations (MaxFEs) in Figure 1. However, in practice we just choose one of the number of evaluations and the number of iterations as the loop termination condition because there is a linear relationship between these two loop termination conditions. In addition, modified contents and added contents are marked in blue font in the new manuscript. With kind regards. (Line 144)
Comment 2:
When we do the benchmarks with other methods, better to provide the citations to those methods, including in the figures and tables. For example, cite the methods in Figure 5. By doing so, it is easier to see which one is the proposed method. Besides, usually the proposed method is located last. Thus, in my opinion, better to put WGWO to the right side of the figure.
- Thanks for your valuable the reminder. Under your reminder, we have used the same comparison method for all the discussions in subsection 4.2. In both subsection 4.2.1, which introduced the comparison methods, and Table 5 we have included references at the comparison algorithms. And we have placed the proposed method on the right side of the figure. In addition, modified contents and added contents are marked in blue font in the new manuscript. With kind regards. (Line 261)
Comment 3:
In Table 6, it is not clear about which figures are being compared. For example, for SSIM, it compares the original image with the segmented image. Why not compare the segmented image with the ground truth? Similarly, for PSNR, how MSE is being calculated?
- Thanks for your valuable the reminder. I will answer the questions you have raised in the following points. First, SSIM and FSIM predicted the perceived image quality and statistically verified the consistency of the segmentation results with those of human observers. PSNR is a full reference image quality metric based on statistical error and correlation. Second, since the threshold segmentation method proposed in this paper deals with unlabeled images, it does not involve experiments for comparison with the ground truth. Third, MSE is obtained from the following formula. , where I denotes the original image, K denotes the segmented image, and the size of both I and K is m×n Since the evaluation metrics used in this paper are common to multi-threshold image segmentation, these metrics are not explained in detail in this paper. But for ease of reading we have introduced references. With kind regards.
Comment 4:
Are there the ground truth? If yes, better to include the ground truth in Figure 5, Figure A1, Figure A2 and Figure A3.
- Thanks for your valuable the reminder. Our proposed thresholding method is based on the pixel information of the original image for image segmentation, so there is no ground truth. With kind regards.
Reviewer 3 Report
The paper provides a well-presented approach for facial region segmentation based on the proposed meta-heuristic framework. The utilization of the Cosmic Wormhole Strategy Enhanced Gray Wolf Optimizer enhances the methodology's novelty and effectiveness. The validation of the proposed method using the IEEE CEC 2020 dataset is meticulously conducted, adding credibility to the research. The organization and execution of the study are commendable, and the results indicate a promising advancement in the field of face image segmentation.
Author Response
[The comments of Reviewer #3]
Comment 1:
The paper provides a well-presented approach for facial region segmentation based on the proposed meta-heuristic framework. The utilization of the Cosmic Wormhole Strategy Enhanced Gray Wolf Optimizer enhances the methodology's novelty and effectiveness. The validation of the proposed method using the IEEE CEC 2020 dataset is meticulously conducted, adding credibility to the research. The organization and execution of the study are commendable, and the results indicate a promising advancement in the field of face image segmentation.
- Thanks for your inspiring comments! We will be working on this for a long time from a methodological and applied point of view.
Reviewer 4 Report
The paper suggests a grey wolf optimization algorithm based on the cosmic wormhole strategy. The proposed method is relatively classic especially when we see that the new segmentation method are based on deep learning approach and have given great results.
In addition, the proposed has been evaluated in a small number images. Besides, the proposed takes much time compared to the existing methods and it seems also slow compared to the recent deep learning based methods.
I did not check the plagiarism. However the English writing is good enough.
Author Response
[The comments of Reviewer #4]
Comment 1:
The paper suggests a grey wolf optimization algorithm based on the cosmic wormhole strategy. The proposed method is relatively classic especially when we see that the new segmentation method are based on deep learning approach and have given great results.
- Thanks for your valuable the reminder. The segmentation method proposed in this paper is different from deep learning based image segmentation methods. This paper focuses more with the study of low-level features of an image and proposes an accurate and stable determination criterion for threshold segmentation. Deep learning, on the other hand, is more based on the known or assumed shape of the target, and uses template matching, deformation modeling, etc., to divide the image into regions that conform to the shape a priori.. In addition, modified contents and added contents are marked in blue font in the new manuscript. With kind regards.
Comment 2:
In addition, the proposed has been evaluated in a small number images. Besides, the proposed takes much time compared to the existing methods and it seems also slow compared to the recent deep learning based methods.
- Thanks for your valuable the reminder. The algorithm complexity segmentation and time consumption experiments show that the proposed method is within the normal processing time of such methods. In addition, sacrificing part of the time consumption to significantly improve the segmentation results is acceptable, and the proposed method in this study has better segmentation quality compared to similar methods. Moreover, the method proposed in this paper eliminates the training process compared to deep learning methods and utilizes the pixel information of the image for direct segmentation. However, I think the problem you pointed out is very valuable. We have similarly considered the problem you have pointed out and have mentioned it in our future work. In the next step of our research, we plan to introduce high-performance computing and parallelism into the segmentation framework and test it on a device with better computational performance. With kind regards.
Reviewer 5 Report
The manuscript proposed an image segmentation method based on the multi-thresholding for face recognition. The detailed description should be added about previous image segmentation methods for face recognition. The image segmentation performance of the proposed method should be compared not only to multi-threshold based methods, but also to state-of-the-art deep learning-based methods.
- (Line 38) The authors mentioned that image segmentation is an important factor in face recognition. Please describe how image segmentation is utilized in face recognition.
- (Line 47) The lack or low quality of training data mentioned as a weakness of deep learning-based image segmentation does not apply to face recognition, i.e. there are many available public datasets for face recognition. In addition, the authors also need to consider an image segmentation network that is trained by unlabeled data, that is, an unsupervised image segmentation. (W-Net: A Deep Model for Fully Unsupervised Image Segmentation, arXiv:1711.08506; Unsupervised Learning of Image Segmentation Based on Differentiable Feature Clustering, arXiv:2007.09990)
- In experiment results, the proposed method was compared with only previous multi-threshold based methods. The performance should be required to be compared with the state-of-the-art deep learning based methods.
- The experimental results in Table 7-9 showed the evaluation metrics for all images and all threshold levels, which are excessively detailed. Table 10-12 sufficiently summarizes them.
Minor editing of English language required
Author Response
[The comments of Reviewer #5]
Comment 1:
(Line 38) The authors mentioned that image segmentation is an important factor in face recognition. Please describe how image segmentation is utilized in face recognition.
- Thanks for your valuable the reminder. Under your reminder, we have discussed the research related to the application of image segmentation techniques for face recognition. In addition, modified contents and added contents are marked in blue font in the new manuscript. With kind regards. (Lines 41 to 47)
Comment 2:
(Line 47) The lack or low quality of training data mentioned as a weakness of deep learning-based image segmentation does not apply to face recognition, i.e. there are many available public datasets for face recognition. In addition, the authors also need to consider an image segmentation network that is trained by unlabeled data, that is, an unsupervised image segmentation. (W-Net: A Deep Model for Fully Unsupervised Image Segmentation, arXiv:1711.08506; Unsupervised Learning of Image Segmentation Based on Differentiable Feature Clustering, arXiv:2007.09990).
- Thanks for your valuable the reminder. Under your reminder, we discussed the problems you pointed out and the methods of unsupervised learning. In addition, modified contents and added contents are marked in blue font in the new manuscript. With kind regards. (Lines 51 to 57)
Comment 3:
In experiment results, the proposed method was compared with only previous multi-threshold based methods. The performance should be required to be compared with the state-of-the-art deep learning based methods.
- Thanks for your valuable the reminder. I agree with you that this is important for a more comprehensive evaluation of the segmentation performance of our proposed model. However, this study is more committed to solving the problem of low quality of image segmentation caused by the difficulty of determining the optimal threshold value in the multi-threshold segmentation process through optimization techniques. In addition, we lacked the time and experience to find segmentation methods on deep learning relevant to this study and reproduce them. Nevertheless, in order to comprehensively evaluate the performance of the proposed model in this paper, we support the feasibility of our proposed method by testing it on benchmark functions as well as image segmentation experiments in which multiple metrics are examined together. In future studies, we will work on the effects of deep learning based segmentation method comparison with your prompting. With kind regards.
Comment 4:
The experimental results in Table 7-9 showed the evaluation metrics for all images and all threshold levels, which are excessively detailed. Table 10-12 sufficiently summarizes them.
- Thanks for your valuable the reminder. Under your reminder, we have revisited the results of this part of the experiment. In addition, modified contents and added contents are marked in blue font in the new manuscript. With kind regards. (Lines 267 to 280)
Round 2
Reviewer 1 Report
In this work, a multi-threshold image segmentation framework is proposed. The proposed framework is based on meta-heuristic optimization method and Kapur's entropy. The meta-heuristic optimization method was optimized to optimize the Kapur's entropy of the image.
In the revised manuscript, the authors have addressed the raised comments.
Author Response
Comment 1:
In this work, a multi-threshold image segmentation framework is proposed. The proposed framework is based on meta-heuristic optimization method and Kapur's entropy. The meta-heuristic optimization method was optimized to optimize the Kapur's entropy of the image.
In the revised manuscript, the authors have addressed the raised comments.
- Thanks for your inspiring comments! We will be working on this for a long time from a methodological and applied point of view.
Reviewer 5 Report
(Line 54-56) Same sentence repeated. In addition, this description is too brief. The disadvantages of the unsupervised image segmentation need to be described in detail.
Minor editing of English language required
Author Response
Comment 1:
(Line 54-56) Same sentence repeated. In addition, this description is too brief. The disadvantages of the unsupervised image segmentation need to be described in detail.
- Thanks for your valuable the reminder. Under your reminder, we have revisited the discussion on segmentation methods for unsupervised learning in Introduction. In addition, modified contents and added contents are marked in blue font in the new manuscript. With kind regards. (Lines 54 to 62)